# Empowering Networks With Scale and Rotation Equivariance Using A Similarity Convolution

**Zikai Sun and Thierry Blu**
Department of Electronic Engineering, The Chinese University of Hong Kong
zksun@link.cuhk.edu.hk, thierry.blu@m4x.org

## Abstract

The translational equivariant nature of Convolutional Neural Networks (CNNs) is a reason for its great success in computer vision. However, networks do not enjoy more general equivariance properties such as rotation or scaling, ultimately limiting their generalization performance. To address this limitation, we devise a method that endows CNNs with simultaneous equivariance with respect to translation, rotation, and scaling. Our approach defines a convolution-like operation and ensures equivariance based on our proposed scalable Fourier-Argand representation. The method maintains similar efficiency as a traditional network and hardly introduces any additional learnable parameters, since it does not face the computational issue that often occurs in group-convolution operators. We validate the efficacy of our approach in the image classification task, demonstrating its robustness and the generalization ability to both scaled and rotated inputs.

## 1 Introduction

The remarkable success of network architectures can be largely attributed to the availability of large datasets and a large number of parameters, enabling them to "remember" vast amounts of information. On the contrary, humans can learn new concepts with very little data and are able to generalize this knowledge. This disparity is due, in part, to the current limitations in modeling geometric deformations in network architectures. Networks are inclined to "remember" data through filter parameters rather than "learning" a full generalization ability. For instance, in classification tasks, networks trained on datasets with specific object sizes often fail when tested on objects with different sizes that were not present in the training set. The ability to factor out transformations, such as rotation or scaling, in the learning process remains to be addressed. It is indeed quite frequent to deal with images in which objects have a different orientation and scale than in the training set, for instance, as a result of distance and orientation changes of the camera.

To mitigate this issue, it is common practice to perform data augmentation (Krizhevsky et al., 2012) prior to training. However, this leads to a substantially larger dataset and makes training more complicated. Moreover, this strategy tends to learn a group of duplicates of almost the same filters, which often requires more learnable parameters to achieve competitive performance. A visualization of the weights of the first layer (Zeiler & Fergus, 2014) highlights that many filters are similar but rotated and scaled versions of a common prototype, which results in significantly more redundancy.

The concept of equivariance emerged as a potential solution to this issue. Simply put, equivariance requires that if a given input undergoes a specific geometric transformation, the resulting output feature from the network (with weights randomly initialized) should exhibit a similarly predictable geometric transformation. Should a network satisfy equivariance to scalings and rotations, training it with only one size and orientation would naturally generalize its performance to all sizes and orientations.

To achieve this property, group convolution methods have been widely used in this field. An oversimplified interpretation of a typical group convolution method is as follows: Features are convolved with dilated filters of the same template to obtain multi-channel features, where the distortion of input features corresponds to the cycle shift between channels. For example, equivariant

CNNs on truncated directions (Cohen & Welling, 2016; Zhou et al., 2017) leverage several directional filters to obtain equivariance within a discrete group. Further works extended rotation equivariance to a continuous group, using techniques such as steerable filters (Weiler et al., 2018; Cohen et al., 2019), B-spline interpolation (Bekkers, 2020) or Lie Group Theory (Bekkers, 2020; Finzi et al., 2020). A similar path to scaling equivariance has been explored, although scaling is no longer intrinsically periodic. The deep scale space (Worrall & Welling, 2019) defined a semi-symmetry group to approximately achieve scale equivariance, while Sosnovik et al. (2020) applied steerable CNNs to scaling. However, integrating equivariance to both rotations and scalings simultaneously leads to a larger group (e.g., a rotation group with $M$ points and a scaling group with $N$ points results in $M \times N$ points for the joint rotation and scaling group), making the task more challenging. Additionally, certain "weight-sharing" techniques based on group convolution can be computationally and memory-intensive.

Despite the difficulties, empowering the model with equivariance of rotation and scaling together can be highly advantageous. For instance, in object detection, the distance changes between the camera and the object, or the random rotations of the object, can significantly impact the accuracy of the method.

The aim of this paper is to propose a CNN architecture that achieves continuous equivariance with respect to both rotation and scaling, thereby filling a void in the field of equivariant methods. To accomplish this, we provide a theoretical framework and analysis that guarantees the network preserves this inherent property. Based on this, we propose a new architecture, the Scale and Rotation Equivariant Network (SREN), that avoids the abovementioned limitations and does not sizably increase computational complexity. Specifically, we first designed a scalable Fourier-Argand representation. The expression of the basis makes it possible to operate the angle

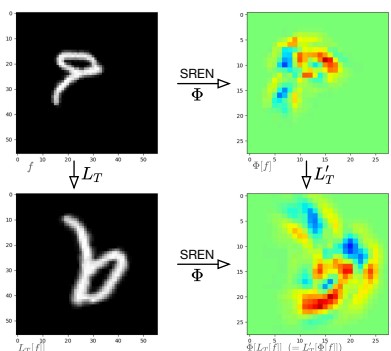

Figure 1: The visualization of the Sim(2) equivariance property: Our SREN method inherently retains the structure information of the input, enabling it to handle distorted objects (rotation, scaling, and translation) without additional training.

and scale in one shot. Further, we propose a new convolution-like operator that is functionally similar to traditional convolution but with slight differences. We show that this new method has similar computational complexity to convolution and can easily replace the typical network structures. Our approach can model both rotation and scaling, enabling it to consistently achieve accurate results when tested with datasets that have undergone different transformations, such as rotation and scaling.

The main contributions in this paper are summarized as follows:

- We introduce the scalable Fourier-Argand representation, which enables achieving equivariance on rotation and scaling.
- We propose the SimConv operator, along with the scalable Fourier-Argand filter, forming the Scale and Rotation Equivariant Network (SREN) architecture.
- SREN is an equivariant network for rotation and scaling that is distinct from group-convolutional neural networks, offering a new possible path to solving this problem for the community.

## 2 RELATED WORK

**Group convolution**    To accomplish equivariance property, a possible direction is the application of group theory to achieve equivariance. Cohen & Welling (2016) introduced group convolution, which enforces equivariance to a small and discrete group of transformations, i.e., rotations by multiples of 90 degrees. Subsequent efforts have aimed to generalize equivariance (Zhou et al., 2017) and also focus on continuous groups coupled with the idea of steerable filters (Cohen & Welling, 2017). To achieve this purpose, Lie group theory has also been utilized, as presented in works such as LieConv (Finzi et al., 2020), albeit only for compact groups. Unfortunately, the scaling group is non-compact, so methods typically treat it as a semi-group and use approximations to achieve truncated scaling equivariance. TridentNet (Li et al., 2019) gets scale invariance by sharing weights among kernels with different dilation rates. Another approach is to apply scale-space theory (Lindeberg, 2013), which

considers the moving band-limits caused by rescaling, as proposed by Worrall & Welling (2019) to attain semi-group equivariance. Bekkers (2020) generate the G-CNNs for arbitrary Lie groups by B-spline basis functions, and can achieve scale or rotation equivariance when using different settings. SiamSE (Sosnovik et al., 2021c) equip the Siamese network with additional built-in scale equivariance. Although Sosnovik et al. (2021b;a) replacing weight sharing scheme with dilated filters can parallel the process, and thus O(1) in terms of time, the overall computational load is still increased concerning the group size. Beyond all these methods, expanding the group to a larger space can increase the overhead computation.

**Steerable filters**    Steerable filters have been used in previous works to achieve equivariance. H-Net (Worrall et al., 2017) and SESN (Sosnovik et al., 2020) are examples of methods that utilized steerable filters to achieve equivariance. Specifically, H-Net utilized complex circular harmonics as filter bases while SESN used Hermite polynomials. The underlying idea behind these methods is to represent filters of different sizes or rotation angles as linear combinations of fixed basis functions. Although this idea is related to our method, achieving rotation-scaling equivariance using this approach is difficult due to the lack of a suitable basis in image processing. Other approaches have also improved equivariance in different aspects. For example, polar transformer networks (Esteves et al., 2018) generalized group-equivariance to rotation and dilation, while attentive group convolutions (Romero et al., 2020) utilized the attention mechanism to generalize group convolution. Some other techniques, such as Shen et al. (2020); Jenner & Weiler (2022), proposed using partial differential operators to maintain equivariance. Additionally, Gao et al. (2022) presented a roto-scale-translation equivariant CNN, but it expanded the filters of the G-CNN in the scale dimension with a truncated interval.

**Differences between related works and ours.**    Unlike previous methods that achieve equivariance only in one aspect, our goal is to ensure continuous rotation and scaling equivariance in a single network. Methods that modify the filters used in the neural network to achieve equivariance require more learnable parameters and are computationally expensive. In contrast, we propose a scalable and steerable filter representation (scalable Fourier-Argand) to modify the convolution operator (SimConv) so that it embodies scale, rotation, and shift equivariance. This approach does not introduce new learnable parameters and allows us to achieve rotation and scale equivariance more efficiently than other methods.

## 3    PRELIMINARIES AND NOTATION

This section clarifies the notations about the sim(2) transformation and the convolution that is often mentioned later. The property of equivariant and invariant are also formulated explicitly, which is derived from our proposed method.

### 3.1    SIM(2) TRANSFORMATION

In Euclidean geometry, two objects can be transformed into each other by a similarity transformation if they share the same shape. This concept of similarity is critical in instance-level computer vision, as objects remain unchanged under scale, translation, and rotation transformations. Consequently, we are motivated to investigate the notion of similarity equivariance. To this end, we introduce the Sim(2) group, which is defined by an invertible translation matrix, as follows:

$$\mathbf{T} = s \begin{bmatrix} \mathbf{R}^\top & -s^{-1}\mathbf{R}^\top \mathbf{t} \\ \mathbf{0} & s^{-1} \end{bmatrix}, \text{and} \quad \mathbf{T}^{-1} = s^{-1} \begin{bmatrix} \mathbf{R} & \mathbf{t} \\ \mathbf{0} & s \end{bmatrix} \in \text{Sim}(2) \subset \mathbb{R}^{3 \times 3} \tag{1}$$

where $\mathbf{R} = \begin{bmatrix} \cos\theta & \sin\theta \\ -\sin\theta & \cos\theta \end{bmatrix} \in \mathbb{R}^{2 \times 2}$ denotes the rotation matrix. The translation vector is denoted by $\mathbf{t} \in \mathbb{R}^2$, and $s \in \mathbb{R}$ is the scale factor. By composing rotation (by an angle $\theta$), scaling (by a factor $s$), and translation (by a vector $\mathbf{t}$) into a single matrix, we obtain a shape-preserving transformation that belongs to the Sim(2) group, given by $\mathbf{T} = \mathbf{A}_s \mathbf{Y}_\mathbf{t} \mathbf{R}_\theta$ Now let's consider a spatial index $\tilde{x} = [x_1, x_2]^\top \in \mathbb{R}^2$ on a 2D image plane. We can extend this index to a 3D homogeneous coordinate by adding a 1 as the third component, that is, $\mathbf{x} = [\tilde{x}, 1]^\top \in \mathbb{R}^3$. Then, for an input signal $f$, we define a linear transformation $L_\mathbf{T} : \mathbb{L}_2(X) \to \mathbb{L}_2(X)$ that transforms feature maps $f \in \mathbb{L}_2(X)$ on some space $X$ according to the Sim(2) transformation $\mathbf{T}$, written as:

$$L_\mathbf{T}[f](\mathbf{x}) = f(\mathbf{T}^{-1}\mathbf{x}) \tag{2}$$

The equation above can be interpreted in the following way: The left-hand side represents a linear transformation $L_{\mathbf{T}}$ applied to a set of feature maps $f$, and the right-hand side represents the value of the feature map $f$ evaluated at the point $\mathbf{T}^{-1}\mathbf{x}$. Moreover, we have $L_{\mathbf{T}}[\cdot] = L_{(\mathbf{A}s\mathbf{Yt}\mathbf{R}\theta)}[\cdot] = L\mathbf{A}s[L\mathbf{Y_t}[L_{\mathbf{R}\theta}[\cdot]]]$, which means that $L_{\mathbf{T}}$ can be decomposed into a series of shape-preserving transformations. The formula for the transformed feature map becomes

$$f(\mathbf{T}^{-1}\mathbf{x}) = f\left([s^{-1}(\mathbf{R}\tilde{x} + 1\mathbf{t})^{\top}, 1]^{\top}\right) \tag{3}$$

This corresponds to a rotation transformation followed by a translation and scaling.

## 3.2 FORMULATION OF CONVOLUTION

Let's consider a stack of two-dimensional features as a function $f(\cdot) = \{f_c(\cdot)\}_{c=1}^{N} : \mathbb{R}^2 \times 1 \to \mathbb{R}^N$, where $N$ is the number of channels. Similarly, a filter bank containing $M$ elements in a convolutional layer can be formalized as $\varphi = \{\psi_k\}_{k=1}^{M}$, where each filter with $N$ channels can be written as $\psi_k = \{\psi_{k,c}\}_{c=1}^{N}$, and $\psi_k : \mathbb{R}^2 \times 1 \to \mathbb{R}^N$ is a vector-valued output filter. Then, for a continuous input with N channels, we regard the spatial cross-correlation between the input and the continuous filter bank $\Psi$ as an operator $\Phi[\cdot] = [\cdot \star \varphi] : \mathbb{R}^N \to \mathbb{R}^M$, written as follows:

$$\Phi[f](\mathbf{x}) = [f \star \varphi](\mathbf{x}) = \{\sum_{c=1}^{N} \int_R f_c(\mathbf{x} - \mathbf{t})\psi_{c,k}(\mathbf{t})\mathrm{d}\mathbf{t}\}_{k=1}^{M} \tag{4}$$

Without loss of generality, we can set $N = M = 1$ and simplify the above formula with the following formula:

$$\Phi[f](\mathbf{x}) = [f \star \varphi](\mathbf{x}) = \int_R f(\mathbf{x} - \mathbf{t})\psi(\mathbf{t})\mathrm{d}\mathbf{t} \tag{5}$$

To explicate, despite the fact that $\mathbf{t}$ is a three-dimensional vector, the integration is still a double integral, along the first and second dimensions, akin to the conventional convolution. The last dimension of $\mathbf{t}$ acts only as a symbolic index, and serves as a mere placeholder. This simplified formulation is employed throughout the remainder of the paper to enhance the comprehensibility of the deduction.

## 3.3 EQUIVARIANCE AND INVARIANCE PROPERTY

**Definition 3.1** (Equivariance). *An operator $\Phi : \mathbb{L}_2(N) \to \mathbb{L}_2(M)$ is equivariant to the transform $L_{\mathbf{T}}$ if there exists a predictable transform $\tilde{L}_{\mathbf{T}}$, such that the following equation holds for any $\mathbf{x} \in \mathbb{R}^{d+1}$.*

$$\Phi[L_{\mathbf{T}}[f]](\mathbf{x}) = \tilde{L}_{\mathbf{T}}[\Phi[f]](\mathbf{x}) \tag{6}$$

If $L_{\mathbf{T}} = \tilde{L}_{\mathbf{T}}$, we can also say that these two operators are commutable. This equivariance property provides structure-preserving properties for the network.

**Definition 3.2** (Invariance). *An operator $\Phi : \mathbb{L}_2(N) \to \mathbb{L}_2(M)$ is invariant to the transform $L_{\mathbf{T}}$ if the following equation holds for any $\mathbf{x} \in \mathbb{R}^{d+1}$.*

$$\Phi[L_{\mathbf{T}}[f]](\mathbf{x}) = \Phi[f](\mathbf{x}) \tag{7}$$

This paper aims to design a convolution-like operation that is similar in function to convolution without lifting the intermediate variable size of the network (as is common in group methods) as well as satisfies the equivariance or invariance property for any similar transform.

## 4 METHOD

In this section, we explore the feasibility of achieving simultaneous equivariance in rotation, scale, and translation for traditional convolutional neural networks. The underlying intuition behind the method is: we first obtain the local scale and orientation of the image. Then, this local information is used to adapt the scale and direction of the filter used for the convolution. Moreover, this (image-dependent) spatially-varying convolution has an efficient implementation.

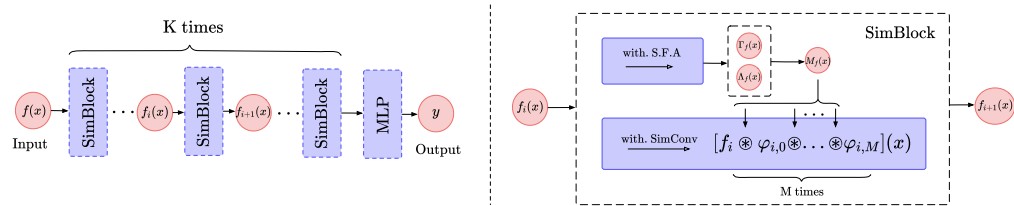

Figure 2: SREN Architecture Overview: The architecture comprises multiple SimBlocks, each utilizing the scalable Fourier-Argand filter (Section 4.1) to extract geometry information $\mathbf{M}_f(\mathbf{x})$. The similarity convolution structure (Section 4.2) combines this indicator to achieve similarity equivariance. A head layer is added to convert the equivariant output to invariant output (Section 4.3).

## 4.1 SCALABLE FOURIER-ARGAND REPRESENTATION

We propose a representation called the Scalable Fourier-Argand Representation to retrieve local geometric information and use it as a covariance indicator in Equation (12). It is exact and steerable. This representation can be defined as follows:

**Definition 4.1** (Scalable Fourier-Argand representation). *Let $h$ be a square-integrable function expressed as $h(r, \theta)$ in polar coordinates. Its scalable Fourier-Argand representation is defined as the following series form:*

$$h(r,\theta) = \sum_{k_1,k_2 \in \mathbb{Z}} \left( h_{k_1,k_2} r^{m_{k_1}} \exp\left( i(k_1\theta + k_2 \frac{2\pi \ln r}{\ln b/a}) \right) \right) = \sum_{k_1,k_2 \in \mathbb{Z}} H_{k_1,k_2}(r,\theta) \qquad (8)$$

*where $\mathbb{Z}$ denotes the set of all integers, and $m_k$ can be chosen as a constant value. The function has limited support, i.e., $(r,\theta) \in [0, 2\pi[ \times [a, b[$, and $k_1, k_2$ can be truncated for a subset of integers to approximate $h$.*

To compute the coefficient of each item $h_{k_1,k_2}$, we can use the following proposition:

**Proposition 4.1.** *Let $h_{k_1,k_2} : \mathbb{Z}^2 \to \mathbb{C}$ as the coefficient of each item. We can compute it as follows,*

$$h_{k_1,k_2} = \frac{1}{\ln \frac{b}{a}} \int_{\ln a}^{\ln b} \frac{1}{2\pi} \int_0^{2\pi} h(e^\rho, \theta) \exp\left( -ik_1\theta - ik_2 \frac{2\pi\rho}{\ln b/a} - \rho m \right) d\theta d\rho \qquad (9)$$

This formula represents the filter as a composition of elementary feature basis. The part about $\theta$ can be seen as the Fourier series of filters on the Argand plane, inspired by Zhao & Blu (2020). The filter itself is related to the harmonic filter (Worrall et al., 2017) in a sense. For the scaling part, we use the logarithmic method to make the scaling of the size a linear shift. We also restrict the support of the function to a plane where the radius ranges from $a$ to $b$, where $a \to 0^+$. In practice, we set $m_k = -1$, which acts like a window function and ensures that the function vanishes at infinity. This is why we call it the Scalable Fourier-Argand Representation. The proof of the proposition can be found in Appendix A.

One benefit of our proposed expression is that its basis functions are steerable for rotation and scalable for scaling. Specifically, consider a transformation matrix $\mathbf{T} = \mathbf{A}_s \mathbf{Y_0} \mathbf{R}_\alpha$, which only involves rotation and scaling. Using this matrix, we have the following equation:

$$\begin{aligned} L_{\mathbf{T}}[H_{k_1,k_2}](r,\theta) &= h_{k_1,k_2} \cdot \exp(ik_1(\theta - \alpha) + ik_2 \frac{2\pi \ln r/s}{\ln b/a}) \cdot (rs^{-1})^{m_{k_1}} \\ &= H_{k_1,k_2}(r,\theta) \cdot \exp(-ik_1\alpha - ik_2 \frac{2\pi \ln s}{\ln b/a}) s^{-m_{k_1}} \end{aligned} \qquad (10)$$

This equation holds for any $\alpha \in [0, 2\pi)$ and $s \in (a, b)$. The following proposition is easily verified:

**Proposition 4.2.** *For a continuous filter $h \in \mathbb{R}^2$ that can be decomposed by a set of basis, let $\mathbf{T} = \mathbf{A}_s \mathbf{Y_0} \mathbf{R}_\alpha$ as a transformation matrix. Then the transformed filter of $h$, noted as $L_{\mathbf{T}}[h]$, can*

*still be represented by the same basis, with a steerable linear combination:*

$$
\begin{aligned}
L_{\mathbf{T}}[h](r, \theta) &= \sum_{k_1, k_2} L_{\mathbf{T}}[H_{k_1, k_2}](r, \theta) \\
&= \sum_{k_1, k_2} H_{k_1, k_2}(r, \theta) \cdot \left( \exp(-ik_1\alpha - ik_2 \frac{2\pi \ln s}{\ln b/a}) s^{-m_{k_1}} \right)
\end{aligned}
\tag{11}
$$

Using this property, we can keep the basis $H_{k_1, k_2}$ fixed and estimate the filter $h$ for different rotations and scales by using different linear combinations. Additionally, to convolve the input signal $f$ with $h$ and its various $L_{\mathbf{T}}[h]$, we can pre-convolve the image with the basis $H_{k_1, k_2}$ of the original filter. Moreover, to achieve more robust results than traditional convolution, we use the normalized cross-correlation denoted as $\star$. This allows us to calculate the intermediate variable $f_{k_1, k_2}$ using the following formula:

$$
f_{k_1, k_2}(\mathbf{x}) = [f \star H_{k_1, k_2}](\mathbf{x}) = \frac{[f * H_{k_1, k_2}](\mathbf{x}) - \mu_i(\mathbf{x})\mu_{H_{k_1, k_2}}}{\sigma_i(\mathbf{x})\sigma_{H_{k_1, k_2}}}
\tag{12}
$$

where $*$ denotes convolution, while $\mu$ and $\sigma$ present the mean and variance of the signal or basis filters. By using these bases, we can determine the optimal orientation and scale through the calculation of the argmax, expressed as follows:

$$
[\Lambda_f(\mathbf{x}), \Gamma_f(\mathbf{x})] = \arg\max_{(\lambda, \gamma)} \sum_{k_1, k_2 = -K}^{K} f_{k_1, k_2}(\mathbf{x}) \cdot c_{k_1, k_2}(\lambda, \gamma) = \arg\max_{\gamma, \lambda} \mathbf{c}_{\gamma, \lambda}\mathbf{F}
\tag{13}
$$

Where $c_{k_1, k_2}(\lambda, \gamma) = \exp(-ik_1\gamma - ik_2 \frac{2\pi \ln \lambda}{\ln b/a})\lambda^{-m}$ is a coefficient that only relys on $\lambda$ and $\gamma$. $\mathbf{c}_{\gamma, \lambda}$ and $\mathbf{F}$ are vectors of all possible $f_{k_1, k_2}$ and $c_{k_1, k_2}$ $\Lambda_f(\mathbf{x}), \Gamma_f(\mathbf{x})$ can be understood as the projection of signal $f$ for the orientation and scale aspect. These two indicators meet the following properties,

**Lemma 4.1.** *Let $\mathbf{T} = \mathbf{A}_s \mathbf{Y}_t \mathbf{R}_\alpha$ as a similarity transformation. Then for a input image $f$ and its distorted version $L_{\mathbf{T}}[f](\mathbf{x})$, for any position $x$, we can have a relationship of this pair of images by following property,*

$$
\begin{aligned}
\Lambda_{L_{\mathbf{T}}[f]}(\mathbf{x}) &= \Lambda_f(\mathbf{T}^{-1}\mathbf{x}) \cdot s \\
\Gamma_{L_{\mathbf{T}}[f]}(\mathbf{x}) &= \Gamma_f(\mathbf{T}^{-1}\mathbf{x}) + \alpha
\end{aligned}
\tag{14}
$$

This is the condition we achieved and applied in Section 4.2. The proof can be found in Appendix B.

## 4.2 SIMILARITY CONVOLUTION

We propose SimConv, a new convolution-like operation that serves as an alternative to traditional convolution. Our design aims to meet the following criteria: 1). It should exhibit equivariance properties for rotation, scale, and translation theoretically. 2). It should incorporate learnable parameters and extract image features by "blending" one function with another. 3). It should have a computational complexity similar to that of traditional convolution, avoiding the computational disaster problem faced by group convolution when extending the group size. These criteria drive us design the SimConv as follows,

**Definition 4.2** (Similarity convolution). *The similarity convolution between the input signal $f$ and the filter $\varphi$ is defined as*

$$
[f \circledast \varphi](\mathbf{x}) := \int_R f(\mathbf{x} + \mathbf{M}_f(\mathbf{x})\mathbf{t})\varphi(\mathbf{t})d\mathbf{t} = \Phi[f](\mathbf{x})
\tag{15}
$$

*where $\mathbf{M}_f(\mathbf{x})$ is a pixel-wise matrix defined as:*

$$
\mathbf{M}_f(\mathbf{x}) = \mathbf{A}_{(\Lambda_f(\mathbf{x}))}\mathbf{R}_{(\Gamma_f(\mathbf{x}))} \in \text{Sim}(2)
\tag{16}
$$

The SimConv operator exhibits a convolution-like structure between the feature map $f$ and the learnable filter $\varphi$. This becomes apparent when we substitute the variable with $\mathbf{t} = \mathbf{M}_f^{-1}(\mathbf{x})\tilde{\mathbf{t}}$. Moreover, it can be degenerate to traditional convolution when replace $\mathbf{M}_f(\mathbf{x})$ to the identity matrix for all $\mathbf{x}$. Furthermore, we define the SimConv as an operator $\Phi[\cdot] = [\cdot \circledast \varphi] : \mathbb{R}^N \to \mathbb{R}^M$, where

we set $M = N = 1$ for readability. This operator is, in fact, equivariant for rotation, scaling, and translation, which we can verify below when considering $\mathbf{M}_f(\mathbf{x})$ in Equation (16) and combining it with the condition in Equation (35).

$$
\begin{aligned}
\mathbf{M}_{L_{\mathbf{T}}[f]}(\mathbf{x})\mathbf{T}^{-1} &= \mathbf{A}_{(\Lambda_{L_{\mathbf{T}}[f]}(\mathbf{x}))}\mathbf{R}_{(\Gamma_{L_{\mathbf{T}}[f]}(\mathbf{x}))}\mathbf{T}^{-1} \\
&= \mathbf{A}_{(\Lambda_f(\mathbf{T}^{-1}\mathbf{x}))}\mathbf{A}_s\mathbf{R}_{(\Gamma_f(\mathbf{T}^{-1}\mathbf{x}))}\mathbf{R}_\alpha\mathbf{T}^{-1} \\
&= \mathbf{A}_{(\Lambda_f(\mathbf{T}^{-1}\mathbf{x}))}\mathbf{R}_{(\Gamma_f(\mathbf{T}^{-1}\mathbf{x}))} = \mathbf{M}_f(\mathbf{T}^{-1}\mathbf{x})
\end{aligned}
\tag{17}
$$

This leads to the following condition: $\mathbf{T}^{-1} = \mathbf{M}_{L_{\mathbf{T}}[f]}^{-1}(\mathbf{x}) \cdot \mathbf{M}_f(\mathbf{T}^{-1}\mathbf{x})$ ,$\forall \mathbf{x} \in \mathbb{R}^2 \times 1$. If we apply a similarity transformation $L_{\mathbf{T}}$, which rotates the signal by $\alpha$ degrees centered at $t$ and scales it by $s$, to the signal $f$ following Equation (2), we obtain the following transformation of the SimConv output:

$$
\begin{aligned}
\Phi[L_{\mathbf{T}}[f]](\mathbf{x}) = [L_{\mathbf{T}}[f] \circledast \varphi](\mathbf{x}) &= \int_R L_{\mathbf{T}}[f](\mathbf{x} + \mathbf{M}_{L_{\mathbf{T}}[f]}(\mathbf{x})\mathbf{t})\varphi(\mathbf{t})\mathrm{d}\mathbf{t} \\
&= \int_R f(\mathbf{T}^{-1}(\mathbf{x} + \mathbf{M}_{L_{\mathbf{T}}[f]}(\mathbf{x})\mathbf{t}))\varphi(\mathbf{t})\mathrm{d}\mathbf{t} = \int_R f(\mathbf{T}^{-1}\mathbf{x} + \mathbf{T}^{-1}\mathbf{M}_{L_{\mathbf{T}}[f]}(\mathbf{x})\mathbf{t})\varphi(\mathbf{t})\mathrm{d}\mathbf{t}
\end{aligned}
\tag{18}
$$

Furthermore, we can deduce the commutator operator of the above as

$$
\begin{aligned}
L_{\mathbf{T}}[\Phi[f]](\mathbf{x}) = L_{\mathbf{T}}[[f \circledast \varphi]](\mathbf{x}) &= L_{\mathbf{T}}[\int_R f(\mathbf{x} + \mathbf{M}_f(\mathbf{x})\mathbf{t})\varphi(\mathbf{t})\mathrm{d}\mathbf{t}] \\
&= \int_R f(\mathbf{T}^{-1}\mathbf{x} + \mathbf{M}_f(\mathbf{T}^{-1}\mathbf{x})\mathbf{t})\varphi(\mathbf{t})\mathrm{d}\mathbf{t} = \int_R f(\mathbf{T}^{-1}\mathbf{x} + \mathbf{T}^{-1}\mathbf{M}_{L_{\mathbf{T}}[f]}(\mathbf{x})\mathbf{t})\varphi(\mathbf{t})\mathrm{d}\mathbf{t}
\end{aligned}
\tag{19}
$$

By replacing the second $\mathbf{T}$ in Equation (19) and comparing it with Equation (18), we have,

$$
\Phi[L_{\mathbf{T}}[f]](\mathbf{x}) = L_{\mathbf{T}}[\Phi[f]](\mathbf{x})
\tag{20}
$$

This implies that applying a similarity transform to the input signal $f$ and then applying the SimConv is equivalent to first applying the SimConv and then the transform, which means that the proposed SimConv satisfies the equivalence property when using the scalable Fourier Argand representation. Moreover, if each layer in the network satisfies this property, the transformation can be propagated from the first layer to the last layer. This can be expressed using the following formula:

$$
f_n(\mathbf{x}) = [L_{\mathbf{T}}[f_0] \circledast \varphi_0 \circledast ... \circledast \varphi_n](\mathbf{x}) = [L_{\mathbf{T}}[f_0 \circledast \varphi_0 \circledast ... \circledast \varphi_n]](\mathbf{x})
\tag{21}
$$

There are several ways to convert equivariant features into invariant features. One option is to add an adaptive max pooling layer at the end of the network. Let $P[\cdot] = \text{torch.nn.AdaptiveMaxPool2d(1)}$ be a function that takes the maximum response over the entire spatial domain. Since the maximum value is not affected by the position distortion of the feature, the output is invariant. Using Equation (20), we obtain:

$$
P[\Phi[L_{\mathbf{T}}[f]]](\mathbf{x}) = P[L_{\mathbf{T}}[\Phi[f]]](\mathbf{x}) = P[\Phi[f]](\mathbf{x})
\tag{22}
$$

We can see that the transformation matrix $\mathbf{T}$ has no influence on the output, which can be useful in tasks such as classification.

## 4.3 DISCRETIZATION METHOD

Although the continuous formulation in the previous section is necessary to go, from the intuitive approach (find scale + orientation, then filter accordingly), to the efficiently implementable formulation (Equation (35)) through a change of variables in an integral, digital images or feature maps are usually discrete data aligned on a mesh grid. Therefore, in this section, we provide a detailed description of the discretization of the integral and its approximation implementation. We rewrite Equation (15) in discrete form as follows:

$$
\Phi[f](x) = [f \circledast \varphi](\mathbf{x}) = \frac{V}{n} \sum_{t \in \mathcal{R}} f(\mathbf{x} + \mathbf{M}_f(\mathbf{x})\mathbf{t})\varphi(\mathbf{t})
\tag{23}
$$

where $\mathcal{R}$ is the support of $\varphi$ and $n$ is the number of elements in the set $\mathcal{R}$. For example, in the case of a $3 \times 3$ convolution, $\mathcal{R} = \{(t_x, t_y, 1)^T | t_x, t_y \in \{-1, 0, 1\}\}$. We define $\mathbf{y_t} = \mathbf{x} + \mathbf{M}_f(\mathbf{x})t$, which is generally a fractional location index. We approximate $f(\mathbf{y_t})$ using bilinear interpolation, written as

$f(\mathbf{y_t}) = \sum_{\mathbf{m}} G(\mathbf{y_t}, \mathbf{m}) f(\mathbf{m})$, where $\mathbf{m} = (m_1, m_2, 1)^T \in \mathbb{Z}^3$ and $G$ is the bilinear interpolation kernel defined as $G(m, n) = g(m_1, n_1) g(m_2, n_2)$, where $g(a, b) = max(0, 1 - |a - b|)$. Thus, the similarity convolution becomes:

$$\Phi[f](\mathbf{x}) = [f \circledast \varphi](\mathbf{x}) = \frac{V}{n} \sum_{\mathbf{t} \in \mathcal{R}} \sum_{\mathbf{m}} G(\mathbf{y_t}, \mathbf{m}) f(\mathbf{m}) \varphi(\mathbf{t}) \qquad (24)$$

With this implementation, we make the similarity convolution differentiable. The gradient of the input is:

$$\frac{\partial \Phi[f](\mathbf{x})}{\partial \mathbf{x}} = \frac{V}{n} \sum_{\mathbf{t} \in \mathcal{R}} \sum_{\mathbf{m}} \frac{\partial G(\mathbf{y_t}, \mathbf{m})}{\partial \mathbf{x}} f(\mathbf{m}) \varphi(\mathbf{t}) \qquad (25)$$

Since $G$ is a differentiable function, and it is non-zero only when $m$ aligns on the grid of $y_t$, it does not require too much computation.

## 5 EXPERIMENTS

### 5.1 CHARACTER RECOGNITION TASK

**Dataset.** The MNIST-ROT-12K DATASET (Larochelle et al., 2007) is commonly used to evaluate rotation-equivariant algorithms. However, it is inadequate to test the algorithms' equivariance on both scaling and rotation. To facilitate fair model comparisons, we construct the SRT-MNIST DATASET by modifying the original MNIST dataset in a similar way. Specifically, we pad the images to $56 \times 56$ pixels, keep the training set unchanged, and randomly apply rotations, scalings, and translations to each test image within the ranges of $\theta = [0, 2\pi)$, $s = [1, 2[$, and $t = \pm 10$. This out-of-distribution setting of the test images provides a sufficient evaluation of the model's generalization ability.

**Experimental setup.** We adopt ResNet-18 He et al. (2016) as the baseline architecture and replace every convolutional layer with our proposed SimConv while retaining the same trainable parameters. The scalable Fourier-Argand filters are shared across all layers. We use Adam optimizer (Kingma & Ba, 2015) with a weight decay of 0.01, initialize the weights with Xavier (Glorot & Bengio, 2010), and set the learning rate to 0.01, which decays by a factor of 0.1 every 50 epochs. We set the batch size to 128 and stop training after 200 epochs. The model has 11.68M parameters, and the FLOPs for an input image of $56 \times 56$ pixels is 0.12G.

Table 1: Generalization ability test on SRT-MNIST

| Methods | Type of the test set. | | | |
|---|---|---|---|---|
| | MNIST | R-MNIST | S-MNIST | SRT-MNIST |
| CNNs | 99.46 | 44.41 | 73.21 | 33.56 |
| SO(2)-Conv | 99.23 | **97.18** | 72.85 | 70.72 |
| $\mathbb{R}^*$-Conv | 99.31 | 35.23 | **99.21** | 39.2 |
| SREN | 99.12 | 96.91 | 98.48 | **92.3** |
| SREN+ | 99.42 | 98.3 | 99.28 | 95.1 |

**Generalization ability study.** We ablate our method's equivariance on rotation and scaling separately and concurrently using SRT-MNIST, where R-MNIST and S-MNIST are datasets with only rotation or scaling applied to test images. We compare SREN with SO(2)-Conv and $\mathbb{R}^*$-Conv, which have the same structure but lack equivariance partially by setting all $\Gamma_f$ or $\lambda_f$ to unitary. Additionally, we include SREN+ with random image rotation and scaling by $\pm 30$ degrees and $[0.8, 1.2]$, respectively. From Table 1, our SREN algorithm achieves an accuracy rate of over 95% on every dataset, whereas the CNNs overfit the original dataset with limited generalization ability.

**Equivariance error analysis.** We conduct numerical validation of the method's equivariance using the equivariant error. This helps us assess the stability of the equivariance property and identify the key factor that affects its stability. The equivariant error is defined by measuring the normalized $L - 2$ distance as follows,

$$\text{Error} = \frac{||L_{\mathbf{T}}[\Phi[f]] - \Phi[L_{\mathbf{T}}[f]]||_F^2}{||L_{\mathbf{T}}[\Phi[f]]||_F^2} \qquad (26)$$

where $|| \cdot ||_F$ is the Frobenius norm. The formula is the relative percentage error of the two obtained features after the input is first convolved then transformed, and vice versa. We compare the $k$-th layer feature with the convolutional network. The average error, shown in Figure 3, is below 0.01, indicating that our network achieves high-quality equivariance. Furthermore, we tested a specific case

where the image is rotated by 90 degrees, and the equivariance error is negligible, with a value of $1.73 \times 10^{-6}$. This result demonstrates that our method can achieve extremely accurate equivariance when without discretization issues.

## 5.2 NATURAL IMAGE CLASSIFICATION

**experimental setup.** To evaluate the generalization ability of our method, we conduct experiments on the STL-10 dataset Coates et al. (2011). The labeled subset is

Table 2: Performance and equivariance comparison on the STL-10 dataset.

| Methods | $\mathcal{R}$-Equi | $\mathcal{S}$-Equi | Conti | ID Accuracy (%) | OOD Accuracy (%) |
|---|---|---|---|---|---|
| ResNet-16 | ✗ | ✗ | ✗ | $82.66 \pm 0.53$ | $37.63 \pm 1.95$ |
| RDCF | ✓ | ✗ | ✗ | $83.66 \pm 0.57$ | $51.12 \pm 4.21$ |
| SESN | ✗ | ✓ | ✓ | $83.79 \pm 0.24$ | $47.26 \pm 0.63$ |
| SDCF | ✗ | ✓ | ✗ | $83.83 \pm 0.41$ | $43.60 \pm 0.87$ |
| RST-CNN | ✓ | ✓ | ✗ | $84.08 \pm 0.11$ | $58.31 \pm 3.62$ |
| **SREN** | ✓ | ✓ | ✓ | $\mathbf{85.25 \pm 0.61}$ | $\mathbf{63.42 \pm 2.57}$ |

an excellent choice to evaluate how efficient the network can use these limited training samples and how much the network's generalization ability is when the training set is small. We evaluate the methods using in-distribution testing (ID), which keeps the test dataset unchanged, and out-of-distribution(OOD) testing, which randomly rotates and scales the dataset. The OOD test measures the ability of a method to handle the never-seen inputs. We use a ResNet (He et al., 2016) with 16 layers as the backbone and replace all convolutional layers with our SimConv layers. The network is trained for 1000 epochs with a batch size of 128, using Adam as the optimizer. The initial learning rate is set to 0.1 and adjusted using a cosine annealing schedule during training. Following Zhu et al. (2019), data augmentation without scaling and rotation is also applied. We compare our method with several other methods, including the Rotation Decomposed Convolutional Filters network (RDCF) (Cheng et al., 2019), Scale-Equivariant Steerable Networks (SESN) (Sosnovik et al., 2020), Scale Decomposed Convolutional Filters network (SDCF) (Zhu et al., 2019), and Roto-Scale-Translation Equivariant CNNs (Gao et al., 2022). We choose ResNet with 16 layers as the backbone for all methods to make the model parameters comparable and ensure a fair comparison.

**Results & Discussion.** Table 2 presents our main results compared to recent baselines. The columns $\mathcal{R}$-Equi and $\mathcal{S}$-Equi indicate whether a method achieves equivariance in rotation or scaling, respectively. The column Conti indicates whether a method achieves equivariance at a continuous scale or rotation. Our method achieved the highest accuracy among all other approaches, particularly for the out-of-distribution test, demonstrating its superior generalization ability. It should be noted that we compare our method with those that only"partially" achieve equivariance, as there lacks like-for-like methods (achieving *both* rotation and scaling equivariant in the *continuous* group) to compare. While MacDonald et al. (2022) guarantees equivariance to any finite-dimensional Lie

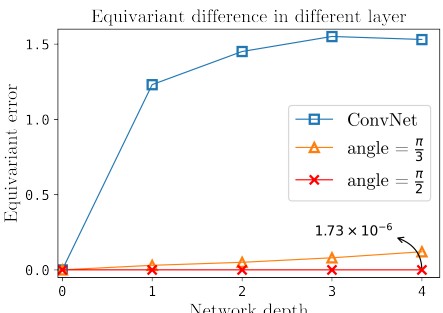

Figure 3: Our method exhibits high equivariant quality in a multi-layer network, and achieving an error level of $10^{-6}$ when there is no discretization approximation.

group, its memory efficiency limits its scalability to large networks and comparison with our method. Achieving equivariance efficiently is challenging, being able to achieve this property itself is already another state-of-the-art.

## 6 CONCLUSION

Although there have been numerous studies on how to achieve rotation and scale equivariant, achieving continuous equivariant in rotation and scaling simultaneously is novel, to our knowledge. In this paper, we propose to develop scalable steerable filters based on the Fourier-Argand representation and to use the local scale and orientation provided by these filters to empower the convolution operator with local scale and rotation equivariant: SimConv. Mathematical and experimental analyses are detailed to explain why it works and to what extent it can achieve the desired property.

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

## A    PROOF OF SCALABLE FOURIER ARGAND REPRESENTATION

**Definition A.1** (Scalable Fourier-Argand representation). *Let $h$ be a square-integrable function expressed as $h(r, \theta)$ in polar coordinates. Its scalable Fourier-Argand representation is defined as the following series form:*

$$h(r, \theta) = \sum_{k_1, k_2 \in \mathbb{Z}} \left( h_{k_1, k_2} r^{m_{k_1}} \exp\left(i(k_1 \theta + k_2 \frac{2\pi \ln r}{\ln b/a})\right)\right) = \sum_{k_1, k_2 \in \mathbb{Z}} H_{k_1, k_2}(r, \theta) \tag{27}$$

*where $\mathbb{Z}$ denotes the set of all integers, and $m_k$ can be chosen as a constant value. The function has limited support, i.e., $(r, \theta) \in [0, 2\pi[ \times [a, b[$, and $k_1, k_2$ can be truncated for a subset of integers to approximate $h$. Let $h_{k_1, k_2} : \mathbb{Z}^2 \to \mathbb{C}$ as the coefficient of each item. We can compute it as follows,*

$$h_{k_1, k_2} = \frac{1}{\ln \frac{b}{a}} \int_{\ln a}^{\ln b} \frac{1}{2\pi} \int_0^{2\pi} h(e^\rho, \theta) \exp\left(-ik_1\theta - ik_2 \frac{2\pi\rho}{\ln b/a} - \rho m\right) d\theta d\rho \tag{28}$$

*Proof.*

We begin by substituting the variable $r$ in Equation (27) with an exponential term, $r = e^\rho$, where $\rho = \ln r$.

$$h(e^\rho, \theta) = \sum_{k_1, k_2 \in \mathbb{Z}} \left[ h_{k_1, k_2} \exp(\rho m_{k_1}) \exp(i(k_1\theta + k_2 \frac{2\pi\rho}{\ln b - \ln a})) \right] \tag{29}$$

Additionally, we introduce the quantity $g_{k_1, k_2}$ and define it accordingly,

$$g_{k_1, k_2} \equiv \frac{1}{\ln b/a} \int_{\ln a}^{\ln b} \frac{1}{2\pi} \int_0^{2\pi} h(e^\rho, \theta) \exp(-ik_1\theta - ik_2 \frac{2\pi\rho}{\ln b/a} - \rho m) d\theta d\rho \tag{30}$$

We then substitute $h(e^\rho, \theta)$ in Equation (30) with the scalable Fourier-Argand representation in Equation (29).

$$
\begin{aligned}
g_{k_1, k_2} &= \frac{1}{\ln b/a} \int_{\ln a}^{\ln b} \frac{1}{2\pi} \int_0^{2\pi} h(e^\rho, \theta) \exp(-ik_1\theta - ik_2 \frac{2\pi\rho}{\ln b/a} - \rho m) d\theta d\rho \\
&= \frac{1}{\ln b/a} \int_{\ln a}^{\ln b} \frac{1}{2\pi} \int_0^{2\pi} \left( \sum_{t_1, t_2 \in \mathbb{Z}} h_{t_1, t_2} \exp(\rho m_{t_1}) \exp(i(t_1\theta + t_2 \frac{2\pi\rho}{\ln b/a})) \right) \\
&\qquad\qquad \exp(-ik_1\theta - ik_2 \frac{2\pi\rho}{\ln b/a} - \rho m) d\theta d\rho \\
&= \frac{1}{\ln b/a} \int_{\ln a}^{\ln b} \frac{1}{2\pi} \int_0^{2\pi} \sum_{t_2 \in \mathbb{Z}} \left( \sum_{t_1 \in \mathbb{Z}} h_{t_1, t_2} \exp(\rho m_{t_1}) \exp(it_1\theta) \right) \exp(it_2 \frac{2\pi\rho}{\ln b/a}) \\
&\qquad\qquad \exp(-ik_1\theta) d\theta \exp(-ik_2 \frac{2\pi\rho}{\ln b/a} - \rho m) d\rho \\
&= \frac{1}{\ln b/a} \int_{\ln a}^{\ln b} \sum_{t_2 \in \mathbb{Z}} \left( \sum_{t_1 \in \mathbb{Z}} h_{t_1, t_2} \frac{1}{2\pi} \int_0^{2\pi} \exp(\rho m_{t_2}) \exp(i(t_1 - k_1)\theta) d\theta \right) \exp(it_2 \frac{2\pi\rho}{\ln b/a}) \\
&\qquad\qquad \exp(-ik_2 \frac{2\pi\rho}{\ln b/a} - \rho m) d\rho \\
&= \frac{1}{\ln b/a} \int_{\ln a}^{\ln b} \sum_{t_2 \in \mathbb{Z}} \sum_{t_1 \in \mathbb{Z}} h_{t_1, t_2} \left( \frac{1}{2\pi} \int_0^{2\pi} \exp(i(t_1 - k_1)\theta) d\theta \right) \\
&\qquad\qquad \exp(it_2 \frac{2\pi\rho}{\ln b/a}) \exp(-ik_2 \frac{2\pi\rho}{\ln b/a}) d\rho
\end{aligned}
\tag{31}
$$

$$
\begin{aligned}
&= \frac{1}{\ln b/a} \int_{\ln a}^{\ln b} \sum_{t_2 \in \mathbb{Z}} \sum_{t_1 \in \mathbb{Z}} h_{t_1,t_2} \delta[t_1 - k_1] \exp(it_2 \frac{2\pi\rho}{\ln b/a}) \exp(-ik_2 \frac{2\pi\rho}{\ln b/a}) d\rho \\
&= \frac{1}{\ln b/a} \int_{\ln a}^{\ln b} \sum_{t_2 \in \mathbb{Z}} h_{k_1,t_2} \exp(i(t_2 - k_2)\frac{2\pi\rho}{\ln b/a}) d\rho \\
&= \sum_{t_2 \in \mathbb{Z}} h_{k_1,t_2} \frac{1}{\ln b/a} \int_{\ln a}^{\ln b} \exp(i(t_2 - k_2)\frac{2\pi\rho}{\ln b/a}) d\rho \\
&= \sum_{t_2 \in \mathbb{Z}} h_{k_1,t_2} \delta[t_2 - k_2] \\
&= h_{k_1,k_2}
\end{aligned}
\tag{32}
$$

The validity of the last equation is demonstrated through two cases: first, when $t_2 = k_2$,

$$
\frac{1}{\ln b/a} \int_{\ln a}^{\ln b} \exp(i(t_2 - k_2)\frac{2\pi\rho}{\ln b/a}) d\rho = \frac{1}{\ln b/a} \int_{\ln a}^{\ln b} \exp(i0) dr = 1
\tag{33}
$$

and second, when $t_2 \neq k_2$.

$$
\frac{1}{\ln b/a} \int_{\ln a}^{\ln b} \exp(i(t_2 - k_2)\frac{2\pi\rho}{\ln b/a}) d\rho = \frac{1}{\ln b/a} \int_{\ln a}^{\ln b} \exp(in\frac{2\pi\rho}{\ln b/a}) d\rho = 0
\tag{34}
$$

Finally, the assertion in Equation (31) is proven by the aforementioned arguments.

## B  PROOF OF DISTORTION PROPERTY

**Lemma B.1.** *Let $\mathbf{T} = \mathbf{A}_s \mathbf{Y}_t \mathbf{R}_\alpha$ as a similarity transformation. Then for a input image $f$ and its distorted version $L_{\mathbf{T}}[f](\mathbf{x})$, for any position $x$, we can have a relationship of this pair of images by following property,*

$$
\begin{aligned}
\Lambda_{L_{\mathbf{T}}[f]}(\mathbf{x}) &= \Lambda_f(\mathbf{T}^{-1}\mathbf{x}) \cdot s \\
\Gamma_{L_{\mathbf{T}}[f]}(\mathbf{x}) &= \Gamma_f(\mathbf{T}^{-1}\mathbf{x}) + \alpha
\end{aligned}
\tag{35}
$$

*Proof.* Consider the original and distorted image $f(x)$ and $L_T[f](x)$, respectively, and use the scalable Fourier Argand basis. We obtain:

$$
\begin{aligned}
\left[\Lambda_{L_{\mathbf{T}}[f]}(\mathbf{x}), \Gamma_{L_{\mathbf{T}}[f]}(\mathbf{x})\right] &= \arg\max_{(\lambda,\gamma)} \sum_{k_1,k_2=-K}^{K} L_{\mathbf{T}}[f]_{k_1,k_2}(\mathbf{x}) \cdot c_{k_1,k_2}(\lambda,\gamma) \\
&= \arg\max_{(\lambda,\gamma)} \sum_{k_1,k_2=-K}^{K} [L_{\mathbf{T}}[f] * H_{k_1,k_2}](\mathbf{x}) \cdot \exp(-ik_1\gamma - ik_2\frac{2\pi \ln \lambda}{\ln b/a})\lambda^{-m}
\end{aligned}
\tag{36}
$$

Substituting $\mathbf{x}$ with $\mathbf{T}^{-1}\mathbf{x}$, we have:

$$
\begin{aligned}
&\left[\Lambda_{L_{\mathbf{T}}[f]}(\mathbf{T}^{-1}\mathbf{x}), \Gamma_{L_{\mathbf{T}}[f]}(\mathbf{T}^{-1}\mathbf{x})\right] \\
&= \arg\max_{(\lambda,\gamma)} \sum_{k_1,k_2=-K}^{K} [f * L_{T^{-1}}[H_{k_1,k_2}]](\mathbf{T}^{-1}\mathbf{x}) \cdot \exp(-ik_1\gamma - ik_2\frac{2\pi \ln \lambda}{\ln b/a})\lambda^{-m}
\end{aligned}
\tag{37}
$$

Besides, from Equation (10), we have:

$$
L_{\mathbf{T}}[H_{k_1,k_2}](r,\theta) = H_{k_1,k_2}(r,\theta) \cdot \left( \exp(-ik_1\alpha - ik_2\frac{2\pi \ln s}{\ln b/a})s^{-m_{k_1}} \right)
\tag{38}
$$

Hence, we can rewrite the previous equation as:

$$
\begin{aligned}
&\left[\Lambda_{L_{\mathbf{T}}[f]}(\mathbf{T}^{-1}\mathbf{x}), \Gamma_{L_{\mathbf{T}}[f]}(\mathbf{T}^{-1}\mathbf{x})\right] \\
&= \arg\max_{(\lambda,\gamma)} \sum_{k_1,k_2=-K}^{K} [f * L_{\mathbf{T}^{-1}}[H_{k_1,k_2}]](\mathbf{T}^{-1}\mathbf{x}) \cdot \exp(-ik_1\gamma - ik_2\frac{2\pi \ln\lambda}{\ln b/a})\lambda^{-m} \\
&= \arg\max_{(\lambda,\gamma)} \sum_{k_1,k_2=-K}^{K} [f * H_{k_1,k_2} \cdot \left(\exp(ik_1\alpha + ik_2\frac{2\pi \ln s}{\ln b/a})s^{m_{k_1}}\right)](\mathbf{T}^{-1}\mathbf{x}) \\
&\qquad\qquad \cdot \exp(-ik_1\gamma - ik_2\frac{2\pi \ln\lambda}{\ln b/a})\lambda^{-m} \\
&= \arg\max_{(\lambda,\gamma)} \sum_{k_1,k_2=-K}^{K} [f * H_{k_1,k_2}](\mathbf{T}^{-1}\mathbf{x}) \cdot \exp(-ik_1(\gamma-\alpha) - ik_2\frac{2\pi \ln(\lambda/s)}{\ln b/a})(\lambda/s)^{-m} \\
&= \arg\max_{(\lambda s,\gamma+\alpha)} \sum_{k_1,k_2=-K}^{K} [f * H_{k_1,k_2}](\mathbf{T}^{-1}\mathbf{x}) \cdot \exp(-ik_1(\gamma) - ik_2\frac{2\pi \ln(\lambda)}{\ln b/a})(\lambda)^{-m}
\end{aligned}
\tag{39}
$$

Finally, we can conclude that,

$$
\begin{aligned}
\Lambda_{L_{\mathbf{T}}[f]}(\mathbf{x}) &= \Lambda_f(\mathbf{T}^{-1}\mathbf{x}) \cdot s \\
\Gamma_{L_{\mathbf{T}}[f]}(\mathbf{x}) &= \Gamma_f(\mathbf{T}^{-1}\mathbf{x}) + \alpha
\end{aligned}
\tag{40}
$$

$\square$

# C  DETAILS, ANALYSIS, AND VISUALIZATION

## C.1  ADDITIONAL STUDIES

**Stability**   We first quantify the deformation stability, and seek to answer in what degree of distortion can the method persist the equivariant (or the generalization ability) compared with original convolution. To achieve this, we evalue our method on the test dataset that rotated and scaled entirely by a certain amount. The MNIST dataset is used in this experiment. The results of this experiment are presented in Figure 4, which shows the accuracy decay with different settings. From the figure, we observe that when the image is rotated significantly beyond what was seen during training, CNNs experience a large drop in accuracy, whereas our method maintains consistently excellent performance. For scale changing tests, our method retains relatively good capability over a wide range of scale changes. However, for extremely large scale changes, our method experiences performance drop due to the limited filter spot and sampling approximation.

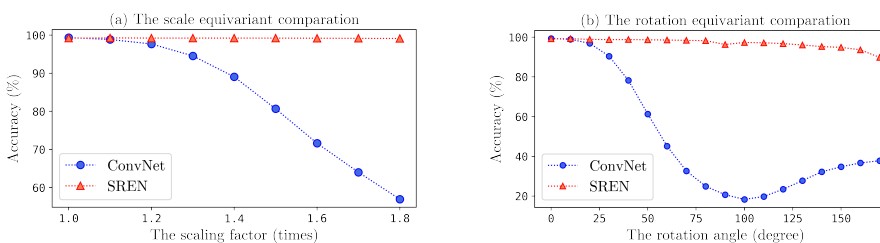

Figure 4: Equivariance Stability: Our network demonstrates high equivariance stability when the entire test set undergoes translation and rotation by a certain scale or angle. In contrast, ConvNet's performance drops significantly under such transformations.

## C.2  FEATURE VISUALIZATION

In addition to numerical experiments, we employ feature visualization to intuitively verify the achieved equivariance of our network. The visualization results are presented in Figure 5. This

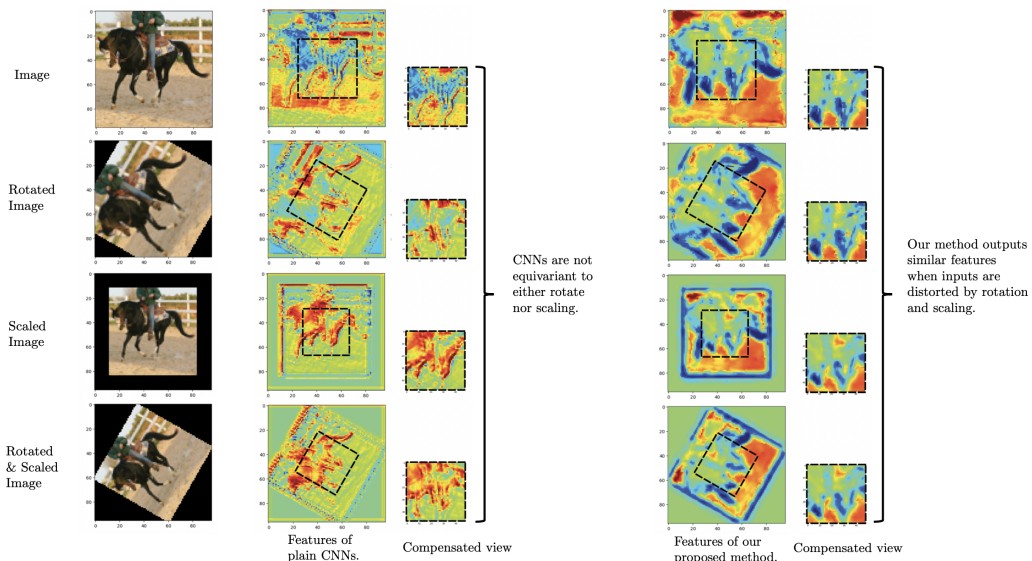

Figure 5: Feature Visualization: We perform a feature visualization of both Convolutional Neural Networks (CNNs) and our Scale-equivariant Residual Equilibrium Network (SREN) using a compensating view. The visualization allows us to observe how the features vary with different transformations of the object. Our results indicate that the features extracted by our method remain stable under transformations, whereas those of CNNs vary.

Table 3: Methods comparison on STL-10 dataset, all methods use the WideResNet as the backbone.

| Method | WRN | SiCNN | SI-ConvNet | DSS | SS-CNN | SESN | DISCO | SREN |
|--------|-----|-------|-----------|-----|--------|------|-------|------|
| Error | 11.48 | 11.62 | 12.48 | 11.28 | 25.47 | 8.51 | 8.07 | 8.23 |

approach examines whether the hidden features of the network exhibit visual equivariance with respect to rotation and scaling. The parameters are randomly initialized for both the CNN and SREN models. Through a compensation view, we can clearly observe that our network successfully achieves equivariance in both rotation and scaling.

## C.3 ARCHITECTURE AND PIPELINE

The algorithmic framework is illustrated in Figure 2. Our method can be extended to a multi-layer network structure. Assuming that the network comprises $K$ blocks (which we named as SimBlock), each of which contains several convolutional layers. Let $f_i$ denote the input signal of the $i$-th block. We first compute the scalable Fourier Argand feature, which is introduced in Section 4.1, to obtain a spatial-wise matrix $\mathbf{M}_{f_i}(\mathbf{x})$. Then, all the SimConv layers, proposed in Section 4.2, within a SimBlock share the same scalable Fourier Argand features. Assuming there are $N$ SimConv layers in each SimBlock, according to the property of Equation (21), the output feature $f_{i+1}$ of the block is guaranteed to be an equivariant feature corresponding to the input image. In the classification task, the invariance property is desired. Therefore, we perform pooling for each channel in the last layer, followed by the MLP layer. Finally, the output is obtained and guaranteed to be invariant for any similar transform. The whole process is presented in Algorithm 1.

## C.4 STL-10 DATASET COMPARATION

This section presents a comparison of our method with other related approaches using the same backbone, namely WideResNet with 16 layers and a widen factor of 8. Our network was trained for 1000 epochs using SGD as the optimizer, with a batch size of 128. The initial learning rate was set to 0.1 and decreased by a factor of 0.2 after 300 epochs, while the drop rate probability was set to 0.3. We also utilized data augmentation techniques, as described in Sosnovik et al. (2021a).

---

**Algorithm 1:** Pipeline of our SREN Algorithm

---

**Input:** Batch of data $\{f_i\}_{i=0}^T$
**Output:** predicted output value $v$
**for** $i = 1 : K$ *levels* **do**
    Initialize filter $h$, calculate its basis $H_{k_1,k_2}$;
    Take $f_i$ as input;
    Get $f_{k_1,k_2} \leftarrow f \star H_{k_1,k_2}$;
    Get $\Gamma_{f_i}, \lambda_{f_i} \leftarrow \arg\max_{\gamma,\lambda} \mathbf{c}_{\gamma,\lambda} \mathbf{F}$;
    Calculate $M_f \leftarrow \Gamma_{f_i}, \lambda_{f_i}$;
    **for** $m = 1 : M$ *layers in specific level n* **do**
        Apply $M_f$ to SimConv;
        $f_{i,m+1} \leftarrow f_i \circledast \varphi_{i,m}$;
    $f_{i+1} \leftarrow f_{i,M}$
Get the output with a head layer: $v \leftarrow \mathrm{MLP}(f)$

---

Table 3 presents our main results compared to recent baseline methods, such as SiCNN (Kanazawa et al., 2014), DSS (Worrall & Welling, 2019), SESN (Sosnovik et al., 2020), and DISCO (Sosnovik et al., 2021a). Our method achieved competitive results that are close to state-of-the-art performance. Additionally, we would like to emphasize that our method's computational cost and parameter count are similar to those of the classic convolutional neural network with the same backbone. This is because the two quantities $\Lambda$ and $\Gamma$ can be shared across different filters, resulting in a small cost compared to the abundance of the convolutional (or our SimConv) layer. Moreover, the complexity of our SimConv operator is similar to that of the convolutional layer. For comparison, as reported in Sosnovik et al. (2021a), the computational cost of the DISCO method takes more than five times longer and SESN takes more than 16.5 times longer than classic CNNs.

