# OpenReview forum: "Empowering Networks With Scale and Rotation Equivariance Using A Similarity Convolution"
_ICLR.cc/2023/Conference — ICLR 2023 poster_

### Official Review · Reviewer_RpmD · 2022-10-24

**Confidence:** 4
**Correctness:** 4
**Technical Novelty And Significance:** 3
**Empirical Novelty And Significance:** 1
**Recommendation:** 8

**Clarity, Quality, Novelty And Reproducibility:**

*Quality:* The approach appears to be technically sound and the claims seem well supported theoretically and empirically.

*Clarity:* although parts of the paper are difficult to read, the paper is generally well organized and the figures are useful.

*Originality:* As far as I know the work is novel.


**Strength And Weaknesses:**

*Strengths*: The authors address the challenging and important problem of designing cnns to produce representations equivariant to larger transformation groups.

*Weaknesses:*
- I found the paper somewhat difficult to read:
  - I believe the paper would benefit from proofreading by an editor.
  - Some additional language in the appendices could make the proofs easier to follow
- I think it is incorrect to say that previous methods can only achieve equivariance to rotation or scaling; the authors themselves cite Esteves et al. 2018 as an example

*Questions:*
- Does the size of the filters impact performance? I ask since (I imagine) the size of the filter determines the scale range.
- I think there may be a typo in Appendix A, I don’t see 4→5 in equation 33, specifically, what happened to exp(\rho m_{m_{t_1}})? Is it possible it should cancel with exp(-\rho m) in 5→6?
- I don’t see 3→4 in Appendix B, should the last quantity be \lambda^{-m}?
- What is meant by a “covariance indicator” (Sec 4.1 para 1)?
- The last step in (19) seems inconsistent with (17) (i.e., pre- vs. post- multiplication with T^{-1}), am I missing something here? Is this because the group is Abelian?
- Is P \circ L_T = P proved somewhere?
- Are \Lambda_f (x), and \Gamma_f (x) computed for every layer at every iteration? Have the authors compared the computational costs of the proposed method to the baseline methods?

*Possible typos:*
- “Specifically, We” → Specifically, we
- ““equivariance in rotation, scale” → equivariant in rotation, scale
- ““if we want to convolute…” → if we want to convolve
- ““We can pre-convolve the filter” → we can pre-convolve the image(?)
- ““implementable formulation (Equation (41))” → implementable formulation (Equation (14))
- ““following summarized condition: T” → following summarized condition: T^{-1}
- ““the communicated operator” → the commutator operator
- ““satisfies the equivalent property” → satisfies the equivariance property
- “Eq 23: “f (x + Mf (x)” → f (x + M_f (x)t)
- ““y_t = x + M_f (x” →y_t = x + M_f (x)t


**Summary Of The Paper:**

The authors propose a method to generate scale-rotation equivariant feature maps in a novel cnn architecture. This is achieved by constraining the filters to be steerable with respect to a scale-rotation equivariant basis. The authors define this basis to be image dependent and validate the approach on STL-10 and variants of MNIST.


**Summary Of The Review:**

I think the proposed work is interesting and of value to the community; however, there are several typos which should be resolved.

---

> ### Author Response · Authors · 2022-11-17
> **Response to Reviewer RpmD**
>
> We thank the reviewer for the detailed and constructive feedback. In addition, we are very grateful to the reviewer for taking the time to carefully check the paper and proofread it.
>
> > **Q1** Does the size of the filters impact performance? I ask since (I imagine) the size of the filter determines the scale range.
>
> Empirically, the geometric information of an instance is of high level and requires a bigger receptive field (or function support). It is appropriate to increase the filter size. Furthermore, because SimConv layers can share the same scalable Fourier-Argand feature, the filter can be increased to a big size without increasing the computational cost too much.
>
>
> > **Q2**  I think there may be a typo in Appendix A, I don’t see 4→5 in equation 33, specifically, what happened to exp(\rho m_{m_{t_1}})? Is it possible it should cancel with exp(-\rho m) in 5→6?
>
> Yes, you are right. We thank the reviewer for pointing out the typo in Appendix 1. $m_{k_1}$ should be $m$ or $m_{k_2}$ and can be cancelled out. We also make efforts to improve the readability of the proof and correct the typo related to $m$ term in appendix A.
>
>
> > **Q3**  I don’t see 3→4 in Appendix B, should the last quantity be \lambda^{-m}?
>
> Yes, there is a typo and it should be $\lambda^{-m}$, we have correct the typo. Thanks.
>
>
> > **Q4**  What is meant by a “covariance indicator” (Sec 4.1 para 1)?
>
> The covariance indicator to which we refer is Equation 14 ($\Lambda$ and $\Gamma$). When a transformation $T$ is applied to a feature $f$, the indicator map of $\Lambda$ and $\Gamma$ not only shifts spatially, but its value changes depending on the degree of rescaling and rotation. This property is known as covariance. The insight in constructing this accurate and steerable representation is that we can utilize it to get local geometric information and use it as a covariance indicator in subsequent section. We rephrased this sentence for clearity.
>
>
> > **Q5**  The last step in (19) seems inconsistent with (17) (i.e., pre- vs. post- multiplication with T^{-1}), am I missing something here? Is this because the group is Abelian?
>
> Yes, one can think of it as the Abelian group. Also, the matrix $M$ and $T^{-1}$ can commute for any $x$ because they can both be  factorized to a combination of rotation and scaling matrices, where rotations in plane are commutative and scaling is diagonal matrix and thus commutative. The groups of rotation/scaling matrices are abelian groups under matrix multiplication.
>
> > **Q6**  Is P \circ L_T = P proved somewhere?
>
> In the corrected transcript, we have added a few sentences to explain this and rectify the inaccuracy. This equation holds true for several types of layers. For classification tasks, the adaptive max pooling layer with the setting of output W=H=1 is used. Some more solution such as taking only the center points of the feature for every channel and stretch then into a vector can also achieve this.
>
>
> > **Q7**  Are \Lambda_f (x), and \Gamma_f (x) computed for every layer at every iteration? Have the authors compared the computational costs of the proposed method to the baseline methods?
>
> (1) Not for every layer, but at every iteration. We do not need to compute these quantities for each layer since filters within and between layers can share the same $\Lambda_f (x)$ and $\Gamma_f (x)$. These values are computed for each iteration. However, if the whole network shares the same $\Lambda_f (x)$ and $\Gamma_f (x)$, one can not necessarily compute it for each iteration.
>
> (2) The computation cost of our method and parameters are just similar as the classic convolutional neural network, if the backbone is the same. This is because $\Lambda$ and $\Gamma$ can be shared within different filters. Thus the cost is low when compared to the large number of convolutional (or our SimConv) layers. And the complexity of our SimConv operator is comparable to that of the convolutional layer.

---

### Official Review · Reviewer_TG34 · 2022-10-25

**Confidence:** 4
**Correctness:** 4
**Technical Novelty And Significance:** 4
**Empirical Novelty And Significance:** 3
**Recommendation:** 8

**Clarity, Quality, Novelty And Reproducibility:**

**Clarity:**

The writing is well-paced and clear to me. However, I think the authors should give more intuition rather than math in the main text. And the intuition parts should be before the rigorous mathematical statements parts. Moreover, I think it might help a lot if the authors can design some simplified concrete 1D examples before talking about the general cases.

**Quality:**

The technical parts are solid. Although I am not able to check all the math line-by-line, I believe the main results are all correct. The experimental results are sufficient to support their claim.

**Novelty:**

The way of constructing equivariant networks presented in this paper is new and has the advantages of computational efficiency and being generalizable outside of the group theory framework. In addition, according to the author, no previous work has achieved equivariance w.r.t. continuous translation/rotation/scaling at the same time. This paper offers a very simple solution.

**Reproducibility:**

The paper has provided sufficient technical details. The code has not been provided yet but one should be able to implement their model with the given details.

**Details Of Ethics Concerns:**

I have no ethical concern.

**Strength And Weaknesses:**

**Strength:**

1. The proposed construction of equivariant networks does not rely on group theory, so it can potentially generalize to transformations that are hard to describe with transformation groups. Furthermore, their implementation is efficient and does not have computational overhead for larger groups.
2. The technical parts are solid, and their empirical study sufficiently supports their claim.

**Questions (potential weaknesses):**

1. The modulation of filters depends on the optimal scale/orientation (computed in equation 13) which has the form of argmax function. I think it can lead to computational instability because: Firstly there might be multiple maximum values, how can you choose under this scenario? Secondly, even though this extreme case does not happen in practice, there may be cases where a little bit of noise can lead to dramatic change, e.g. orientation $0^{\circ}$ is only slightly better than $180^{\circ}$, a small amount of noise could lead to the filters rotate by $180^{\circ}$. How are you planning to fix this issue?
2. Still on equation 13, it seems to me that the scale/orientation of the filters depends on the input signal over the entire domain. That is to say, to compute local features, a change far away could change the orientation/scale of the filters. Would that break the locality of the convolutional layers?

However, even with all the potential drawbacks mentioned above, I still think this is an interesting work that can inspire valuable research.



**Summary Of The Paper:**

This paper provides a new way to construct convolution-like neural networks that are equivariant to continuous translation, rotation, and scaling without relying on group theory. More specifically, they use the local scale/orientation of the input signal to adapt the scale/orientation of the filters (with the proposed Fourier-Argand representation) and implement spatially-varying convolutions. The proposed method does not introduce any significant computational overhead. They empirically verify the equivariance and generalization ability of their proposed models on scaled and rotated versions of MNIST and STL-20 datasets.

**Summary Of The Review:**

I recommend acceptance of this paper based on the following:
This paper provides a new way to construct convolution-like neural networks that is equivariant to translation/rotation/scaling, with appealing computational efficiency, and can potentially be generalized to other transformations without relying on the group theory framework. The technical parts are solid and the empirical study is sufficient. Even though I have some technical concerns outlined in the question section, they do not stop this paper from being valuable research work.

---

> ### Author Response · Authors · 2022-11-17
> **Response to Reviewer TG34**
>
>
> We thank the reviewer for the detailed and constructive feedback.
>
> > **Q1** 	The modulation of filters depends on the optimal scale/orientation (computed in equation 13) which has the form of argmax function. I think it can lead to computational instability because: Firstly there might be multiple maximum values, how can you choose under this scenario? Secondly, even though this extreme case does not happen in practice, there may be cases where a little bit of noise can lead to dramatic change, e.g. orientation 0 degree is only slightly better than 180 degree, a small amount of noise could lead to the filters rotate by 180 degree. How are you planning to fix this issue?
>
> Thanks for the great question. Because of the characteristics of numerical calculations, the chance of this extreme cases when two decimals are precisely identical is quite unlikely. However, it is possible that two or even more maximums are quite close to one other. This, however, is not a major issue. The following are the reasons:
>
> Let $f$ as a feature map, and we have two arguments of the similar maxima
> $\gamma^* = \gamma_0 = \gamma_0 + \theta$ at point $\mathbf{x_0}$. From Equation 13, we have,
> $$
> \sum_{k_1, k_2=-K}^{K} f_{k_1, k_2}({\bf x_0}) \cdot c_{k_1, k_2} (\lambda, \gamma_0)
>  -
> \sum_{k_1, k_2=-K}^{K} f_{k_1, k_2}({\bf x_0}) \cdot c_{k_1, k_2} (\lambda, \gamma_0 + \theta)
> \simeq 0
> $$ where $c_{k_1, k_2} (\lambda, \gamma) = \exp (-i k_1 \gamma -i k_2 \frac{ 2 \pi \ln \lambda}{\ln b / a} ) \lambda^{-m}$.
> So we can have,
> $$
> \sum_{k_1, k_2=-K}^{K} \left[ f_{k_1, k_2}({\bf x_0}) \cdot (1-\exp (-i k_1 \theta) ) \right] \cdot \exp (-i k_1 \gamma_0 -i k_2 \frac{ 2 \pi \ln \lambda}{\ln b / a} ) \lambda^{-m}
> \simeq 0
> $$
> This leads to,
>  $$
>  f({\bf x_0}) - L_{\bf R_{\theta}}[f] ({\bf x_0}) \simeq 0
>  $$
> Where $R$ is the rotation matrix with parameter $\theta$.
> In the reviewer's case, $\theta=\pi$ means the local feature itself is centrosymmetric. So even if $\gamma^*$ is unstable, convolving the centrosymmetric feature with a kernel or the same kernel but rotated by $\theta=\pi$ gives the same result. This special case explains what would happen if this case were to occur.
>
> A more realistic scenario is that the local feature only includes a flat constant (for example, the image at this local position is a low-level structure). As a result, all $\lambda$ and $\gamma$ maxima are indistinguishable. However, this local feature convolving with the kernel with arbitrary distortion will provide a similar result. As a result, this instability will have little impact on output, and do not need to pay much attention to fix this issue.
>
>
> >**Q2** 	Still on equation 13, it seems to me that the scale/orientation of the filters depends on the input signal over the entire domain. That is to say, to compute local features, a change far away could change the orientation/scale of the filters. Would that break the locality of the convolutional layers?
>
>
> The filter is still a local filter since there is a $r^m$ term in $H_{k_1, k_2} (r, \theta)$, where we can set the value of $m$ (e.g., $m=-1$) to control the meaningful support of the filter. As a result, a change far away will not affect the local output and would not break the locality.

---

> > ### Comment · Reviewer_TG34 · 2022-11-21
> > **Response to authors**
> >
> > Thanks for the explanation. The authors have addressed my questions. Therefore, I would like to recommend acceptance.

---

### Official Review · Reviewer_MrDN · 2022-10-28

**Confidence:** 4
**Correctness:** 3
**Technical Novelty And Significance:** 3
**Empirical Novelty And Significance:** 3
**Recommendation:** 6

**Clarity, Quality, Novelty And Reproducibility:**

The paper is clear, the illustrations are of good quality. The approach and reasoning are novel. I am not able to evaluate it's reproducibility because the code is not provided. However, from the provided information it is possible to implement the main building blocks of the paper. However, I want to highlight the fact that if the authors release the code, it will improve the visibility of such an approach in the community, which is always a huge contribution!

**Strength And Weaknesses:**

The paper is well-written. Although, I am going to propose several adjustments, the paper is already clear enough to be understood. The paper has a good structure which makes it easy to follow the  idea of the authors. Additionally, the mathematical language is high level which allows to stick to very accurate formulations.

The idea discussed in the paper is novel. And it is interesting to read it. Although it may be rough, and may be at an early stage of its development, it made me start thinking about the area of the field, the problem and the solutions, which authors highlighted. It is a big plus.

The main weakness of the paper is that the novelty sounds very important, it seems to be of a large scale, while the experiments just slightly demonstrate its advantage over the previous methods.

**Issues, which affect my rating**
1. If I understood correctly, the rotation part of the proposed filters is very close to the idea of Harmonic Nets by Worral et al. [1] and E(2)-steerable CNNs with irreducible representations by Cesa & Weiler [2]. If so, then I think that a more accurate comparison to [1, 2] should be presented. If not, please add some remarks to the sections 4.1 to highlight the difference.
2. The authors mention several times that the method is not based on group theory. I think such a distinction is misleading, artificial and somewhat incorrect mathematically. Moreover, the exact roots or each cited paper go far beyond than just "group theory". If it is not based on group theory, could you demonstrate how this fact may be useful to a reader. For example, can it be generalized to a non-group transformation?
3. Introduction, the "To achieve this property," paragraph. I find several phrases here a bit misleading and are of less respect to the previous approaches. I suppose it is just wording. For instance, the phrase "these “copy-paste” approach with group theory" may give a reader an incorrect understanding of the previous appoaches. What the authors call here as "copy-paste" is better to be referred to as "weight-sharing". It is a) more convenient and b) less vulgar. Finally, for some methods, such as [1, 2] it is not necessary to reuse filters to obtain their rotated versions.
4. One of the arguments for the proposed method is that it allows for equivariant models with lower complexity than quadratic. In Worrall & Welling [3] and Sosnovik et al. [4, 5] it is demonstrated that scaling can be implemented as dilation (integer and fractional). Moreover, it is possible to implement it by utilizing multiple cores of GPUs to make it O(1) in terms of time, which is also discussed in the papers, as the weight sharing scheme is replaced with dilation in these papers. I think a more accurate comparison of computational complexities would help.
5. I suppose that the results in Table 5 are copied from the paper [6]. The main issue with equivariant models when it comes to comparing a new model to the previous approaches is the hyperparameter tuning. In the original papers the authors spent some time to perform such a process. For example in [2, 4, 5, 7] the authors train and test by using exactly the same protocol and their reported results are more accurate than what is present in Table 2. One of the main reasons is because they used a WideResNet. If it is possible to additionally compare a WideResNet with the proposed blocks to the previous models. It would significantly increase the contribution of section 5.2

**Issues, which do not affect my rating**
1. The paper contains typos and writing issues. For instance, the word "trannels" (p1) seems rather misleading. The word "convolute" (p5) is incorrectly used (I suppose it must be "convolve").
2. In Eq. 1 it makes sense to write the transformation as a 3x3 Matrix from the very start by using the block notation. It will improve the clarity of the equation. Because then, you introduce the 3rd coordinate to $[x_1, x_2, w]$. It also makes sense to write what the $w$ is.
3. A more detailed illustration of how the equivariance of the SimBlock works would help.
4. Very bottom, page 2. I am not sure if "Worrall and Welling" is the right reference for "Scale-space theory". I suppose the placing of the reference is just incorrect.

- [1] Worrall D. E. et al. Harmonic networks: Deep translation and rotation equivariance. CVPR – 2017
- [2] Weiler M., Cesa G. General E(2)-equivariant steerable CNNs. NeurIPS - 2019.
- [3] Worrall D., Welling M. Deep scale-spaces: Equivariance over scale. NeurIPS – 2019.
- [4] Sosnovik I., Moskalev A., Smeulders A. Disco: accurate discrete scale convolutions. BMVC - 2021
- [5] Sosnovik I., Moskalev A., Smeulders A. How to Transform Kernels for Scale-Convolutions. CVPR – 2021, VIPriors Workshop
- [6] Gao L., Lin G., Zhu W. Deformation robust roto-scale-translation equivariant cnns. TMLR - 2022
- [7] Sosnovik I., Szmaja M., Smeulders A. Scale-equivariant steerable networks. ICLR 2020

**Summary Of The Paper:**

The authors highlight the fact that CNNs, which are capable of tackling translation, scale and rotation transformations, would be useful for computer vision, but are less studied than less general counterparts. Then the authors propose scalable Fourier-Argand representation as the key for building such models. In their experiments they demonstrate that the proposed SREN model is more accurate than the previous methods.

**Summary Of The Review:**

The paper contains a novel idea and highlights an important issue: combining translation, scale and rotation in one networks is still challenging from the implementation point of view.

---

> ### Author Response · Authors · 2022-11-17
> **Response to Reviewer MrDN (1/2)**
>
> We thank the reviewer for the valuable and constructive feedback.
>
> > **Q1** If I understood correctly, the rotation part of the proposed filters is very close to the idea of Harmonic Nets by Worral et al. [1] and E(2)-steerable CNNs with irreducible representations by Cesa & Weiler [2]. If so, then I think that a more accurate comparison to [1, 2] should be presented. If not, please add some remarks to the sections 4.1 to highlight the difference.
>
> **A1** We agree that the proposed representation's basis filter (if just considering the rotation component) is similar to the notion of H-Net or E2CNN (E(2)-steerable CNNs method is a more generic framework that contains Harmonic Nets.). However, even neglecting the scaling part, what we have (a representation) still differs from the Harmonic filter in H-Net. What we offer is a representation of an arbitrary function rather than a steerable filter. This means that by choosing adequate basis coefficients (Equation 9), we ensure that we can have the exact rotated and scaled version of the same original function. This is the reason we call it a representation. This exact property allows us to extract the hidden geometric information to two quantities $\Lambda$ and $\Gamma$. And we use it as a condition to satisfy the equivariant property in our SimConv layer (whereas the harmonic filter itself cannot do it).
>
> The second distinction is that the SimConv layer contains the majority of the learnable filters in our network, and because $\Lambda$ and $\Gamma$ can be shared between layers, the cost is rather low. On the contrary, Harmonic Net  parameterizes the vector spaces of the G-steerable kernel. The steerable kernel also includes the learnable ability. As a result, the entire network is made up of a vast number of steerable filters, rather than spacial convolutional filter as our work did.
>
> However, since section 4.1 (Method section) illustrates the representation itself, which definitely has a link with the harmonic filter, we totally agree to add remarks and discussion here to highlight the connection and difference between our method and others, as suggested by the reviewer.
>
> > **Q2** The authors mention several times that the method is not based on group theory. I think such a distinction is misleading, artificial and somewhat incorrect mathematically. Moreover, the exact roots or each cited paper go far beyond than just "group theory". If it is not based on group theory, could you demonstrate how this fact may be useful to a reader. For example, can it be generalized to a non-group transformation?
>
> **A2** (This question is the same with question 1.1 of Reviewer Ke2s) We thank the reviewer for pointing out our inaccurate method description. We did use a general and vague statement to distinguish us from most methods, which is inappropriate. We have revised the abstract and introduction of the paper. We find the phrase ”the group-convolutional neural networks” or “G-convolutions” more accurate in expressing our statement and distinguishing our method from the group-convolution-based methods. We actually mean that we do not use the famous group convolution ideas to achieve equivariance.
>
> For example, this paper[1] is one of the typical group-convolution style algorithms. It first lifts the input image from $\mathbb{Z}^2$ to function in $G$. And all remaining hidden layers employ the group-convolution on features in the $G$ domain. And the convolutional filter itself is a function with the distorted transformation $u$. As a result, each feature in the output feature set is the input feature convolved with the same filter but wrapped in by different transformations in the desired group.
>
> Many additional studies make use of similar principles. However, as compared to original CNNs, this indicates that the intermediate feature in the neural network has an additional dimension. In [2], for example, the rotation and scaling group is discretized into $N_r$ and $N_s$ points independently. And the feature is therefore a 5D array of the type $\[ M_l, N_r, N_s, H_l, W_l \]$, where $M l$ is the channel number.
>
> On the contrary, our method does not contain any lifting process, and the features and kernels are more similar to the classic CNNs, which its feature are still a 3D array with the shape $\[ M_l, H_l, W_l \]$. At the same time, since the feature map size has no relationship to the size of the group (which is eager to achieve equivariance). And because of the exact steerable nature of scalable Fourier Argand representation. Our method can achieve exact and continuous equivariant on rotation and scaling.
>
> [1] MacDonald, Lachlan E., Sameera Ramasinghe, and Simon Lucey. "Enabling equivariance for arbitrary Lie groups." Proceedings of the IEEE/CVF Conference on Computer Vision and Pattern Recognition. 2022.
>
> [2] Gao, Liyao, Guang Lin, and Wei Zhu. "Deformation robust roto-scale-translation equivariant cnns." arXiv preprint arXiv:2111.10978 (2021).

---

> > ### Author Response · Authors · 2022-11-17
> > **Response to Reviewer MrDN (2/2)**
> >
> > > **Q3**  Introduction, the "To achieve this property," paragraph. I find several phrases here a bit misleading and are of less respect to the previous approaches. I suppose it is just wording. For instance, the phrase "these “copy-paste” approach with group theory" may give a reader an incorrect understanding of the previous appoaches. What the authors call here as "copy-paste" is better to be referred to as "weight-sharing". It is a) more convenient and b) less vulgar. Finally, for some methods, such as [1, 2] it is not necessary to reuse filters to obtain their rotated versions.
> >
> > Thanks to your suggestions, we have revised that paragraph in the introduction and removed the contentious comment, as follows:
> >
> > *However, integrating equivariance to rotations and scalings at the same time results in a bigger group (e.g., a rotation group with $M$ points and a scaling group with $N$ points results in $M \times N$ points for the joint rotation and scaling group), making the task more difficult. Furthermore, certain "weight-sharing" techniques based on group convolution are computationally and memory storage intensive.*
> >
> > > **Q4** One of the arguments for the proposed method is that it allows for equivariant models with lower complexity than quadratic. In Worrall & Welling [3] and Sosnovik et al. [4, 5] it is demonstrated that scaling can be implemented as dilation (integer and fractional). Moreover, it is possible to implement it by utilizing multiple cores of GPUs to make it O(1) in terms of time, which is also discussed in the papers, as the weight sharing scheme is replaced with dilation in these papers. I think a more accurate comparison of computational complexities would help.
> >
> > Although replacing weight sharing scheme with dilated filters allows the process to be parallelized and hence O(1) in terms of time, the overall computing burden is still raised due to the increased group size. For example, In [4, 5], the methods employs a scale convolution with the set of $\{ 1, 2^{1/3}, 2^{2/3}, 2 \}$, which we suppose the computational efficiency limits this. We added a discussion in related work section.
> >
> >  On the other hand, the approach we offer is to overcome the group size constraint on computational complexity. As we state in response to the second question, our method no longer performs convolution on the Group domain, thus making it possible on an extremely large or even continuously infinite group. This is a significant advantage of our method.
> >
> > In term of the computational complexities. The computation cost and parameters of our technique are comparable to those of a standard convolutional neural network with the same backbone. This is due to the fact that the two quantities $\Lambda$ and $\Gamma$ can be shared between filters. As a result, the cost is low when compared to the abundance of the convolutional (or our SimConv) layer. And the SimConv operator's complexity is comparable to that of the convolutional layer.
> >
> > > **Q5** I suppose that the results in Table 5 are copied from the paper [6]. The main issue with equivariant models when it comes to comparing a new model to the previous approaches is the hyperparameter tuning. In the original papers the authors spent some time to perform such a process. For example in [2, 4, 5, 7] the authors train and test by using exactly the same protocol and their reported results are more accurate than what is present in Table 2. One of the main reasons is because they used a WideResNet. If it is possible to additionally compare a WideResNet with the proposed blocks to the previous models. It would significantly increase the contribution of section 5.2
> >
> > As suggested by the reviewer, we conducted additional experiments in Appendix to evaluate our strategy using WideResNet. Table 3 shows that our method can provide a competitive result that is similar to the state-of-the-art method while also being more computationally efficient than other methods.
> >
> > > **Others**  A more detailed illustration of how the equivariance of the SimBlock works would help. Some typos and better phrasing.
> >
> > Thanks for suggestion, we have updated the discription about the SimBlock in the Architecture section of the Appendix. Besides, we are very grateful to the reviewer for pointing out the typos and misleading phrasing. This significantly increased the quality of the paper.
> >
> > [3] Worrall D., Welling M. Deep scale-spaces: Equivariance over scale. NeurIPS – 2019.
> >
> > [4] Sosnovik I., Moskalev A., Smeulders A. Disco: accurate discrete scale convolutions. BMVC - 2021
> >
> > [5] Sosnovik I., Moskalev A., Smeulders A. How to Transform Kernels for Scale-Convolutions. CVPR – 2021, VIPriors Workshop

---

### Official Review · Reviewer_Ke2s · 2022-11-04

**Confidence:** 4
**Correctness:** 3
**Technical Novelty And Significance:** 4
**Empirical Novelty And Significance:** 4
**Recommendation:** 8

**Clarity, Quality, Novelty And Reproducibility:**

The paper is written well and is generally very clear. The central ideas in the paper are indeed novel. I am not very sure about the reproducibility however, and strongly suggest that the authors try to release their code.


**Details Of Ethics Concerns:**

No ethics concerns.

**Strength And Weaknesses:**

Strengths:

1. The ideas in the paper are novel. The scalable Fourier-Argand representation is clever and allows for enabling the required equivariance, given that the convolution operation is defined as shown.

2. Equivariance has also been experimentally verified.

3. Experiments with SRT-MNIST clearly show that the proposed method outperforms methods that do not take into the right equivariance into account.

Weaknesses:

1. I am confused by the authors stating at multiple points that they are describing the first method that achieves rotation and scaling equivariance simultaneously. But the authors themselves describe several works in the related work section that do it. The most recent example I know is from CVPR 2022: Enabling Equivariance for Arbitrary Lie Groups which shows a framework for equivariance to any finite-dimensional Lie group: https://openaccess.thecvf.com/content/CVPR2022/papers/MacDonald_Enabling_Equivariance_for_Arbitrary_Lie_Groups_CVPR_2022_paper.pdf

The authors should definitely make this point clearer in the paper and discuss why these could not be included as baselines for their experiments. I also don't see what the authors mean by their method not being based on group theory.

2. There are some minor very minor issues in the formulation section 3.2. In many places, the equations use dimension $d$, while the text says two-dimensions and that the convolution integral is a double integral.

 3. While the experiment with STL-10 is appreciated, it would be nice if the authors can have another experiment to clearly show that it is because of the joint scale and rotation equivariance that the experiments improve i.e. are the OOD samples OOD because of unseen scale and rotation transformations?

**Summary Of The Paper:**

The paper describes a new convolution operation for neural networks that is simultaneously equivariant to rotations, translations and scaling. The crux of the method is representing the images and filters using in what the authors call the scalable Fourier-Argand representation. With this operation defined, equivariance can be shown theoretically for continuous signals, which also holds empirically. Implementation of the idea via discretization is also discussed.  Experiments on multiple benchmark datasets clearly show that the proposed ideas are better at classification than other equivariant methods, especially when unknown scale, rotation and translations are applied simultaneously at the input.

**Summary Of The Review:**

I think the novel scalable Fourier-Argand representation and how it enables equivariance is very interesting. Experimental results provide validation. There are some weaknesses which I have listed above which I hope the authors can address. Overall, I recommend acceptance.


UPDATE AFTER AUTHOR RESPONSE:

I thank the authors for their response. I think the contributions are clearer now and the authors should have this discussion in the main paper. Having also read the other reviews, I maintain my original score of accepting the paper.

---

> ### Author Response · Authors · 2022-11-17
> **Response to Reviewer Ke2s (1/2)**
>
> We thank the reviewer for the detailed and constructive feedback.
>
>  > **Q1.1** I also don't see what the authors mean by their method not being based on group theory.
>
> **A1.1** We thank the reviewer for pointing out our inaccurate method description. We did use a general and vague statement to distinguish us from most methods, which is inappropriate. We have revised the abstract and introduction of the paper. We find the phrase ”the group-convolutional neural networks” or “G-convolutions” more accurate in expressing our statement and distinguishing our method from the group-convolution-based methods. We actually mean that we do not use the famous group convolution ideas to achieve equivariance.
>
> For example, this paper[1] is one of the typical group-convolution style algorithms. It first lifts the input image from $\mathbb{Z}^2$ to function in $G$. And all remaining hidden layers employ the group-convolution on features in the $G$ domain. And the convolutional filter itself is a function with the distorted transformation $u$. As a result, each feature in the output feature set is the input feature convolved with the same filter but wrapped in by different transformations in the desired group.
>
> Many additional studies make use of similar principles. However, as compared to original CNNs, this indicates that the intermediate feature in the neural network has an additional dimension. In [2], for example, the rotation and scaling group is discretized into $N_r$ and $N_s$ points independently. And the feature is therefore a 5D array of the type $\[ M_l, N_r, N_s, H_l, W_l \]$, where $M_l$ is the channel number.
>
> On the contrary, our method does not contain any lifting process, and the features and kernels are more similar to the classic CNNs, which its feature are still a 3D array with the shape $\[ M_l, H_l, W_l \]$. At the same time, since the feature map size has no relationship to the size of the group (which is eager to achieve equivariance). And because of the exact steerable nature of scalable Fourier Argand representation. Our method can achieve exact and continuous equivariant on rotation and scaling.
>
> [1] MacDonald, Lachlan E., Sameera Ramasinghe, and Simon Lucey. "Enabling equivariance for arbitrary Lie groups." Proceedings of the IEEE/CVF Conference on Computer Vision and Pattern Recognition. 2022.
>
> [2] Gao, Liyao, Guang Lin, and Wei Zhu. "Deformation robust roto-scale-translation equivariant cnns." arXiv preprint arXiv:2111.10978 (2021).

---

> > ### Author Response · Authors · 2022-11-17
> > **Response to Reviewer Ke2s (2/2)**
> >
> > > **Q1.2** I am confused by the authors stating at multiple points that they are describing the first method that achieves rotation and scaling equivariance simultaneously. But the authors themselves describe several works in the related work section that do it. The most recent example I know is from CVPR 2022: Enabling Equivariance for Arbitrary Lie Groups which shows a framework for equivariance to any finite-dimensional Lie group. The authors should definitely make this point clearer in the paper and discuss why these could not be included as baselines for their experiments.
> >
> > **A1.2** We clarify the novelty of our method in Section 5.2 by "achieving *rotation* and *scaling* equivariant in the *continuous* group". Moreover, since the representation itself is exact, the equivariance we obtained is also continuous and exact, and the method efficiency won't limited by the size of the group.
> > However, in [2], it expands filters in the scale dimension with a truncated interval. As a result, what this paper achieved does not contradict our argument. We do, however, appreciate the reviewer's advice on the more conservative writing in our paper and have amended the somewhat misleading words in the conclusion section.
> >
> > As for the paper mentioned by the reviewer[1], this paper suggests using the Metropolis algorithm (MCMC) to estimate the group convolution, which is a great idea. However, the group, like many earlier techniques, should have finite elements in the group. As a result, the equivariant is still not exact. Moreover, as noted by the authors in its discussion section, "with $L$ convolutional layers, and $N$ samples used to approximate each convolution, one needs $N^L$function evaluations in total. Practically, this means very large memory usage at the front end of the forward pass." We suppose this is why the author use a extremely small network (only two convolutional layers and two residual fully connected layers) throughout the paper. Thus adapting it to a more scalable backbone is challenging, and it is not suited to be included as a baseline for STL-10 experiments.
> >
> > > **Q2** There are some minor very minor issues in the formulation section 3.2. In many places, the equations use dimension d, while the text says two-dimensions and that the convolution integral is a double integral.
> >
> > **A2** We thank the reviewer for carefully checking on the typo. $ d$ equals to 2 and means the same thing. We changed all $d$ to 2 for consistency.
> >
> > > **Q3** While the experiment with STL-10 is appreciated, it would be nice if the authors can have another experiment to clearly show that it is because of the joint scale and rotation equivariance that the experiments improve i.e. are the OOD samples OOD because of unseen scale and rotation transformations?
> >
> > **A3** To some extent, we suppose that testing on OOD can have the conclusion that “the OOD samples OOD because of unseen scale and rotation transformations”. To illustrate, when using ResNet to test on the test set, the accuracy is 82.66%; however, when the test set is rotated and scaled, the accuracy rate reduces to a very low at 37.63%. In comparison, using our method to achieve joint scale and rotation equivariance in the network, the accuracy rate on OOD improves by up to 25.79%.

---

### Decision · Program_Chairs · 2023-01-20

**Decision:**

Accept: poster

**Justification For Why Not Higher Score:**

Some concerns about claims (see above); lack of large-scale experiments; writing quality is somewhat lacking.

**Justification For Why Not Lower Score:**

The paper has an original idea that seems to work well on small-scale problems

**Metareview: Summary, Strengths And Weaknesses:**

The paper proposes a new approach to building convolution-like networks that are equivariant to rotation, translation and scale. The idea is to extract a local scale and orientation and use this to filter at that scale/orientation. The reviewers appreciated the novelty of the idea, the experiments validating the improved OOD performance, and experimental validation of low equivariance error.

Some concerns were raised about certain claims made in the original version (e.g. "not based on group theory") and comparison to previous work. I agree with the reviewers that the original paper, and to some degree the current paper still, makes some unsuported claims and is not entirely balanced in discussion previous work. For instance, already the first "Steerable CNNs" paper mentions the idea of decoupling the feature dimension from the size of the group, and indeed all steerable CNNs achieve this, so this is not a novel feature of the current paper. How this is achieved is however novel; since the SimConv is not linear in the input (since it depends on the nonlinear M_f(x)), it is not a steerable convolution.

One additional note is that the work bears some similarity to "Deformable Convolutional Networks" by Dai et al., and to a lesser extent "Spatial Transformer Networks" by Jaderberg et al. It would be nice to discuss these in the related work as well.

Personally, I would have liked to see larger-scale experiments to show that the method really works on important problems. Adding more details on implementation (e.g. in an appendix), and a code release, is also strongly encouraged.

In summary, this paper proposes a novel and promising approach to joint rotation/translation/scale equivariant CNNs. There are some minor issues remaining and we encourage the authors to address them as much as possible. Nevertheless I recommend that the paper be accepted to ICLR.

**Note From Pc:**

if the above contains the word "oral" or "spotlight" please see: "oral" presentation means -> notable-top-5% and "spotlight" means -> notable-top-25%. As stated in our emails, we are disassociating presentation type from AC recommendations